# Temporal Context Reinstatement Drives Episodic-Like Order Memory in Long-Context Language Models

**Mathis Pink** [1]  **Vy Ai Vo** [1 2]  **Qinyuan Wu** [1]  **Jianing Mu** [3]  **Javier Turek** [4]  **Uri Hasson** [5]  **Kenneth A. Norman** [6]
**Sebastian Michelmann** [7]  **Alexander Huth** [3]  **Mariya Toneva** [1]

## Abstract

Human episodic memory supports the retrieval of experiences that unfold over extended timescales, yet the computational mechanisms underlying this ability remain debated due to the limited mechanistic accessibility in long-term memory experiments in humans. Long-context LLMs may offer promising ways to reveal plausible computational mechanisms that drive this type of retrieval. Here, we investigate whether and how LLMs capture the core behavioral signatures of episodic memory via a temporal order memory task. Using a new dataset of human behavior based on memory of a full-length novel, we show that models exhibit the same characteristic distance effect observed in humans on this task. We next apply long-context mechanistic interpretability analyses to uncover how models solve this task, and find that model performance relies on a one-dimensional temporal code that is reinstated during retrieval by a single time-reinstatement attention head. These findings support temporal context reinstatement as an important mechanism for episodic-like temporal-order memory in LLMs, offering new insights into how temporal aspects of long-term episodic memory may be instantiated in both artificial and biological systems.

## 1. Introduction

Episodic memory enables humans to recall specific past experiences as temporally extended events, preserving information about what occurred and when it occurred within the unfolding stream of experience (Tulving, 1972). This capacity supports reasoning over long timescales, allowing past events to be flexibly retrieved and recombined to guide present behavior (Schacter & Addis, 2007), making episodic memory a cornerstone of biological intelligence. Beyond the human brain, episodic memory is also increasingly relevant for artificial systems: as large language models are deployed in settings that require continuous learning and long-term interaction, the ability to retain, retrieve, and temporally organize information over extended periods has become a key desideratum for modern AI systems.

Despite decades of research, the computational mechanisms that give rise to human episodic memory remain actively debated. Progress has been limited by a fundamental constraint: the mechanisms of human memory systems are difficult to probe and causally manipulate. While behavioral experiments and neural measurements have identified signatures of episodic encoding and retrieval (Xue, 2018), they offer limited leverage for isolating the internal computations that support memory over long, naturalistic experiences. As a result, key mechanistic questions—particularly those concerning how temporal structure is recovered during retrieval—remain unresolved.

Large language models (LLMs) provide a promising complementary framework for investigating these questions. In particular, recent work has increasingly interpreted in-context memory through the lens of episodic memory (Gershman et al., 2025; Kolling et al., 2025; Ji-An et al., 2024; Park et al., 2025), drawing connections between transformer architectures and memory mechanisms in the brain (see Sec. 2 for specific differences from our work). At the same time, important differences remain between memory in transformers and in humans. Causally masked self-attention retains effectively lossless access to past representations, whereas human episodic retrieval is inherently lossy and subject to many additional constraints. Despite these differences, a growing body of work has shown that language models align

[1]Max Planck Institute for Software Systems, Saarbrücken, Germany [2]TR Labs, Thomson Reuters, Toronto, Ontario, Canada [3]University of California, Berkeley, Berkeley, California, USA [4]Earth Dynamics AI, Beaverton, Oregon, USA [5]Department of Psychology, Princeton University, Princeton, New Jersey [6]Princeton Neuroscience Institute, Princeton University, Princeton, New Jersey [7]Department of Psychology, New York University, New York City, New York. Correspondence to: Mathis Pink <mpink@mpi-sws.org>.

*Proceedings of the 43ʳᵈ International Conference on Machine Learning*, Seoul, South Korea. PMLR 306, 2026. Copyright 2026 by the author(s).

with human behavior and neural representations across a range of memory-related cognitive functions, including associative learning (Kolling et al., 2025), event segmentation (Michelmann et al., 2025), and more general language comprehension (Toneva & Wehbe, 2019; Schrimpf et al., 2021; Caucheteux & King, 2022; Goldstein et al., 2022; Vo et al., 2023).

Moreover, advances in long-context architectures allow LLMs to process and retain information over extremely long sequences, approaching the temporal scales characteristic of real-world episodic experiences. Crucially, unlike biological systems, LLMs are fully mechanistically transparent: their internal representations and retrieval dynamics can be examined and directly manipulated. Together, these properties position long-context LLMs as a powerful model system for investigating computational hypotheses about episodic memory.

In this work, we leverage long-context LLMs as a mechanistically accessible model to investigate computational mechanisms that can support one central aspect of episodic memory: the ability to recover the temporal structure of past experiences (i.e., temporal order judgments). Specifically, we focus on two competing hypotheses. The first hypothesis proposes that temporal order judgments rely on an explicit temporal position signal that tracks when events occurred (Howard & Kahana, 2002; Temudo et al., 2025; Umbach et al., 2020). The second hypothesis posits that order can be inferred indirectly from semantic information, such as a causal or narrative chain linking events (Antony et al., 2024; Chen & Bornstein, 2024; Radvansky et al., 2014), without requiring an explicit representation of time.

To enable this investigation, we introduce a new dataset of human temporal order judgments derived from a full-length novel and compare human performance to that of long-context LLMs. We show that models exhibit the same distance effect as humans in their behavioral patterns on this task. Leveraging mechanistic interpretability analyses, we then identify the internal processes that give rise to successful retrieval in the models. Our results reveal that performance depends on the reinstatement of a low-dimensional temporal code during retrieval, mediated by a specialized attention head, rather than solely on semantic inference. Together, these findings indicate that reinstating temporal context plays a central role in supporting episodic-like temporal order memory in LLMs, and shed light on how temporal structure in long-term episodic memory may be realized in both artificial and biological systems.

Our main contributions are:

1. A temporal order task for long-context LLMs, together with a new long-range human memory dataset that characterizes the behavioral profile of the task.

2. A methodology for localizing temporal order information and its reinstatement in attention heads of long-context LLMs.

3. The finding that a single attention head reinstates temporal information from the transformer key-value cache, and is causally implicated in temporal order judgment tasks.

4. Finally, we release ICMemory-Kit, a flexible, open-source toolkit to facilitate future research on long-context memory and mechanistic interpretability.

We hope that our work, which develops approaches for mechanistic localization and causal validation of temporal reinstatement in LLMs, can serve as a starting point for future cognitive models of human memory.

## 2. Related Work

**Computational models of episodic memory.** Existing models of episodic memory, such as the context maintenance and retrieval model (Polyn et al., 2009), are confined to specific task designs. LLMs hold potential to eventually enable models of episodic memory in naturalistic tasks. To advance this direction, we investigate a mechanism in LLMs that is related to episodic-memory: the reinstatement of temporal context.

**Links between LLMs and episodic memory.** A growing line of work draws functional and mechanistic parallels between transformer language models and episodic memory computations.

Gershman et al. (2025) framed episodic memory as a key–value memory system and highlighted a functional correspondence between key–value retrieval in episodic memory and self-attention-based encoding and retrieval in causal LLMs. We adapt this framing and view the formation of a value cache as the encoding of memory for later (attention-based) retrieval.

Park et al. (2025) showed that LLM representations can exhibit latent structure in their principal components, aligned with the data-generating process underlying a prompt, and related this structure to hippocampal function. Our analysis is methodologically adjacent to their spectral and principal-component approach, but while they look for representational geometry that captures structure of the underlying environment that generated the prompt, we look for temporal structure aligned with the sequential processing of the prompt, regardless of what the data-generating process is.

Ji-An et al. (2024) argued that contiguity-biased attention patterns supporting in-context learning can be modeled and identified by using a context maintenance and retrieval

(CMR) model of episodic memory (Polyn et al., 2009), suggesting functional similarities between episodic memory and in-context learning. Here, we find that one retrieval head has a special function: to reinstate a time-code associated with a previously encountered token sequence.

**Positional information as temporal information.** From an episodic-memory perspective, positional signals in sequence models can be interpreted as temporal information. Recent work suggests that transformers can form and use positional/temporal representations even without additional positional scaffolds (e.g., RoPE (Su et al., 2024)), including in "NoPE" settings without any explicit positional encoding (Haviv et al., 2022; Kazemnejad et al., 2023; Chi et al., 2023; Zuo et al., 2025; Irie, 2025; Gelberg et al., 2025). In this work, we focus on identifying where such temporal information is localized in the network, how it is reinstated during processing, and testing its causal contribution in a temporal order judgment task. In relation to this line of work, we show in Section 5.2 and in Appendix A.1 that the temporal information analyzed in this paper does not depend on RoPE.

## 3. Natural and long-timescale episodic memory task

We study temporal order judgments in a naturalistic long-form stimulus. The core task is a binary multiple-choice task: given two 50-word segments sampled from a long text, recall which segment appeared earlier in the original text. **Figure** 1 illustrates the encoding–retrieval structure of this memory task. During *encoding*, the long text is processed (i.e. read) sequentially. During later *retrieval*, the participant (human or model) is shown two non-overlapping segments, denoted A and B (A/B labels assigned randomly), and must answer:

> *Did segment A appear before segment B in the text?*

We call this task *Sequence Order Recall Task* (SORT). Based on human experiments, we found that segments of 50 words provide a good trade-off between task difficulty and task duration. We aligned our human and model experiments with this, i.e. each segment consists of exactly 50 consecutive words taken verbatim from the source text. The query presented during the retrieval phase contains the two segments and the binary question. Randomizing the assignment of the two sampled segments to A/B prevents systematic response biases tied to the presentation order to be reflected in performance. For a given segment pair, we define the *distance* $\Delta_w$ as the number of words between the two segments.

### 3.1. Model experiment

For models, we construct a single long-context prompt in which the full source text appears first (the *encoding span*), followed by a short task instruction (see Appendix A.11 for prompts used) and the two 50-word segments (the *retrieval span*). Models then output a binary decision (A-before-B vs. B-before-A). We present each pair in both orders to remove multiple-choice label biases. The setup ensures that any information used to answer the order question must be derived from representations formed while processing the text in-context, rather than from external tools or iterative search over the source. The exact prompts that we used are shown in Appendix A.11.

### 3.2. Human experiment

Books are a natural probe of long-term episodic memory: they are typically read over multiple sessions and days, are a common real-world activity rather than a synthetic task, and provide a fixed ground truth for what could have been encoded. However, there is no publicly available dataset measuring human long-term memory for entire books at scale. We therefore designed an experiment to collect this data from a large sample and will release it publicly. Given the difficulty of recruiting participants to read lengthy books specifically for an experiment, we used a creative recruiting strategy: inviting members of the online reading community *Goodreads* who had recently finished reading one 300-page book that had become part of the public domain in the same year—*The Murder of Roger Ackroyd*. Participants completed an online survey within 30 days of finishing the book. The expected compensation for participation was $12 and the study was approved by the IRB at Princeton University. We provide 990 segment pair samples from 97 participants. Further details about this study are provided in Appendix A.10.

### 3.3. Datasets

We use one naturalistic book stimulus (the primary dataset used to compare human and model behavior and to validate mechanisms) and additional documents for representational analyses (see **Table** 1). For all texts, we removed temporal markers such as enumerations, dates, chapter numbers etc. Across datasets, a *segment* is a contiguous 50-word window that starts at a new sentence.

### 3.4. Models

We evaluate two long-context LLMs: Llama3.1-8b-Instruct and Llama3.1-70b-Instruct (Grattafiori et al., 2024). We choose these models as both models have a context window of 128k tokens that fits the complete novel and the task, rely on grouped query attention without sliding windows, both

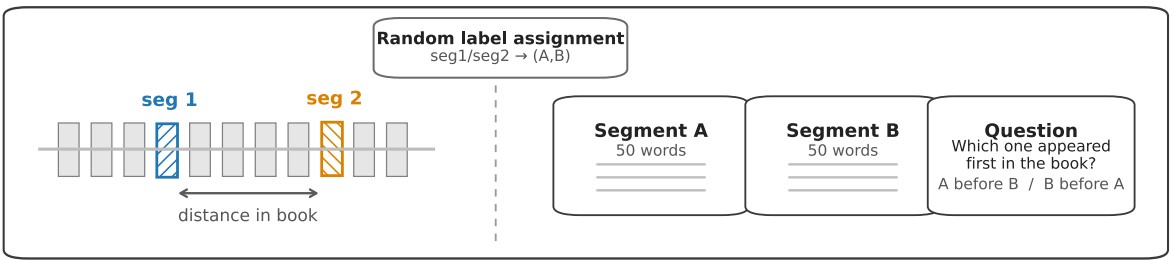

*Figure 1.* **Task schematic.** For each trial, two 50-word segments are sampled from a full-length text with known order and distance in words. At test time they are presented as A/B in random order, and the subject/model answers which segment appeared earlier.

| Document | Purpose | # Pairs | Document length |
|---|---|---|---|
| *TMORA* | behavior causal interventions | 296 | 93k |
| 60k random words | fit readout | 600 | 100k |
| House of Commons (transcript, truncated) | test readout | 600 | 100k |
| *TMORA* (shuffled seeds 1–4) | behavior test readout | 600 | 93k |

*Table 1.* Documents used for temporal order memory tasks, their role in the study, and segment-pair counts. We have human and model data for one novel *The Murder of Roger Ackroyd (TMORA)*, which we use to validate mechanisms and evaluate behavior. 4 different sentence-level shuffled versions of the novel are used to test the degree to which models rely on narrative coherence. A sequence of 60k random English nouns is used to fit linear readout-directions of temporal/positional information. Lastly, to validate the generalization of these temporal readout directions, we use the shuffled versions of the novel, along with a recent transcript of the British Parliament, which we cleaned and truncated to 100k tokens. After identifying a mechanism for time-reinstatement, we validate its causal role on the novel.

are trained in largely the same way, and both are not overly specialized for coding and math problems.

To determine whether the model selects the correct answer, we compare its predicted probabilities for the correct and incorrect labels. For each sample, we evaluate both possible segment-to-label assignments and take an expectation over the label assignment. We mark the response as correct if

$$\mathbb{E}_{a\sim\mathrm{Unif}(\mathcal{A})}\left[p(y_{\mathrm{correct}}\mid a)\right] > \mathbb{E}_{a\sim\mathrm{Unif}(\mathcal{A})}\left[p(y_{\mathrm{incorrect}}\mid a)\right],$$

where $\mathcal{A}$ is the set of the two possible label assignments (see Figure 1).

### 3.5. Statistical tests

For unpaired data, we performed a permutation test to compare the mean performance in each distance bin. Over 20,000 iterations, this test permutes the labels between the two groups of interest (e.g. humans vs. Llama3.1-8B-Instruct) and computes the mean performance of the permuted groups. All comparisons were checked for excess skew and data imbalance, and two-tailed p-values computed by the usual method (Christensen & Zabriskie, 2022). Finally, we applied multiple comparisons correction across distance bins (Benjamini & Yekutieli, 2001).

## 4. Behavioral similarity between LLMs and humans on SORT

Both humans and the models have encoded memory representations of the book during their respective encoding phases. During the retrieval phase, we use pairs of segments from *The Murder of Roger Ackroyd* that are judged by both the humans and models to probe memory of temporal order across different distances (in words) between the segments.

**Distance-dependent performance is aligned between humans and models on *The Murder of Roger Ackroyd*.** Figure 2 shows accuracy as a function of binned distances between segments for humans and for both Llama 3.1-8B-Instruct and Llama 3.1-70B-Instruct evaluated on the same 296 segment pairs from *The Murder of Roger Ackroyd*. Human performance exhibits a clear distance effect: accuracy increases with distance, reaching reliably above-chance performance for widely separated segments while remaining near chance for nearby segments. Both models reproduce this pattern: accuracy increases with distance and reaches $\geq 70\%$ for far-apart segments. The difference between human performance and Llama 3.1-8B-Instruct is not statistically significant (all $p \geq 0.289$). However, Llama 3.1-70B-Instruct performs better than humans for segments between 16k-24k words apart ($p = 0.021$, FDR-corrected; Figure 2).

**Model performance is preserved under sentence-level shuffling.** Distance effects alone do not identify whether temporal order judgments are driven by narrative reconstruction (e.g., causal constraints and plot schemas) or by reinstating a temporally organized trace formed during sequential

processing. To test whether models require narrative structure, we evaluate both models on four shuffled variants of *The Murder of Roger Ackroyd* (with different random seeds), each paired with its own set of 600 segment-pair queries. In each version, the text is shuffled across blocks of four sentences. Sentence-block level shuffling preserves local sentence content but disrupts global narrative coherence and causal dependencies, substantially weakening the relationship between "what happens" and "when it happens" that is present in the original book. Figure 2 shows that performance is largely preserved under shuffling, both models retain a strong distance effect, and achieve comparable accuracy relative to the original *The Murder of Roger Ackroyd* evaluation. For both models, the difference in performance between the intact and shuffled data was not statistically significant after multiple comparisons correction (Llama 3.1-8B-Instruct: all $p \geq 0.307$; Llama 3.1-70B-Instruct: all $p \geq 0.104$). Since collecting human data for a full length shuffled book is infeasible, we do not compare these results to a human reference.

The results indicate that the models' ability to judge temporal order in this setup does not primarily depend on narrative coherence or causal schemas in the text. Instead, the persistence of distance-dependent accuracy under shuffling is consistent with a mechanism that relies on the encoding of temporal positional information during sequential processing in the encoding phase, and later reinstatement of this temporal information during the retrieval-phase.

# 5. Mechanisms for temporal reinstatement in LLMs

Successful episodic memory recall of a segment requires recovery not only of its content but also of its position in the original sequence. For unfamiliar types of sequences, such positional information needs to be retrieved from in-context memory within attention layers. To localize temporal information, we thus analyze head-specific representations of segments across encoding and retrieval, using ground-truth segment positions to assess whether the representational structure is organized primarily by sequential position. We then show how a temporal code formed during the encoding-phase is reinstated during retrieval to drive temporal order memory performance in the models.

To evaluate whether an attention head's representational geometry captures temporal order, we look at the top principal components. PCA has previously been used to localize order information by Dai et al. (2024), and it has been used by Park et al. (2025) to recover data-generating graph-structure in representations. We validate our PCA-based findings against linear probing of positional information in Appendix A.2, and in Appendix A.3, we suggest that using PCA to identify order information is related to spectral seriation

(Fogel et al., 2016).

## 5.1. Representational temporal order judgments

For each dataset, we extract an ordered collection of $N$ segments $\{s_i\}_{i=1}^N$ in *ground-truth* order (increasing text position). For a fixed transformer layer $\ell$ and attention head $h$, we denote the per-token value vectors associated with a segment by

$$v_t^{(\ell,h)} \in \mathbb{R}^{d_h}, \qquad t \in \{i_{\text{enc}}, \ldots, i_{\text{enc}} + T - 1\}, \quad (1)$$

where $T$ is the segment length (in tokens) and $d_h$ is the per-head dimensionality. Likewise, we denote the retrieval-time representations (which are weighted averages over all past value vectors) by

$$o_t^{(\ell,h)} \in \mathbb{R}^{d_h}, \qquad t \in \{i_{\text{ret}}, \ldots, i_{\text{ret}} + T - 1\}, \quad (2)$$

For each segment starting at index $i$, $\mathcal{T}_i^{\text{enc}} = \{i, \ldots, i + T - 1\}$ are the token positions corresponding to the *encoding-phase occurrence* of segment $s_i$ in the long text, and let $\mathcal{T}_i^{\text{ret}}$ be the token positions corresponding to the *retrieval-phase occurrence* of that same segment when it is presented in the retrieval-phase of the prompt. We define segment-averaged representations:

$$r_{\text{enc}}^{(\ell,h)}(i) = \frac{1}{|\mathcal{T}_i^{\text{enc}}|} \sum_{t \in \mathcal{T}_i^{\text{enc}}} v_t^{(\ell,h)} \in \mathbb{R}^{d_h}, \qquad (3)$$

$$r_{\text{ret}}^{(\ell,h)}(i) = \frac{1}{|\mathcal{T}_i^{\text{ret}}|} \sum_{t \in \mathcal{T}_i^{\text{ret}}} o_t^{(\ell,h)} \in \mathbb{R}^{d_h}. \qquad (4)$$

Deduplicating and then stacking these vectors in *ground-truth temporal order* yields the encoding- and retrieval-phase representational data matrices $R_{\text{enc}}$ and $R_{\text{ret}}$:

$$R_{(\cdot)}^{(\ell,h)} = \begin{bmatrix} r_{(\cdot)}^{(\ell,h)}(1)^\top \\ r_{(\cdot)}^{(\ell,h)}(2)^\top \\ \vdots \\ r_{(\cdot)}^{(\ell,h)}(N)^\top \end{bmatrix} \in \mathbb{R}^{N \times d_h}, \qquad (5)$$

## 5.2. Localization of temporal information

Following the linear representation hypothesis (Park et al., 2024), we seek a *linear* direction $w \in \mathbb{R}^{d_h}$ such that the 1D projection scores of the temporally ordered representation matrix $z = Rw$, admit an ordering aligned with $(1, 2, \ldots, N)$.

Given a fixed $w$, we quantify temporal alignment by rank correlation (Kendall's $\tau$) between the sorted indices $\hat{\pi}$ of the projected representations and the ground-truth indices:

$$\hat{\tau}(R; w) = \tau_{\text{K}}(\hat{\pi}(Rw), (1, 2, \ldots, N)). \qquad (6)$$

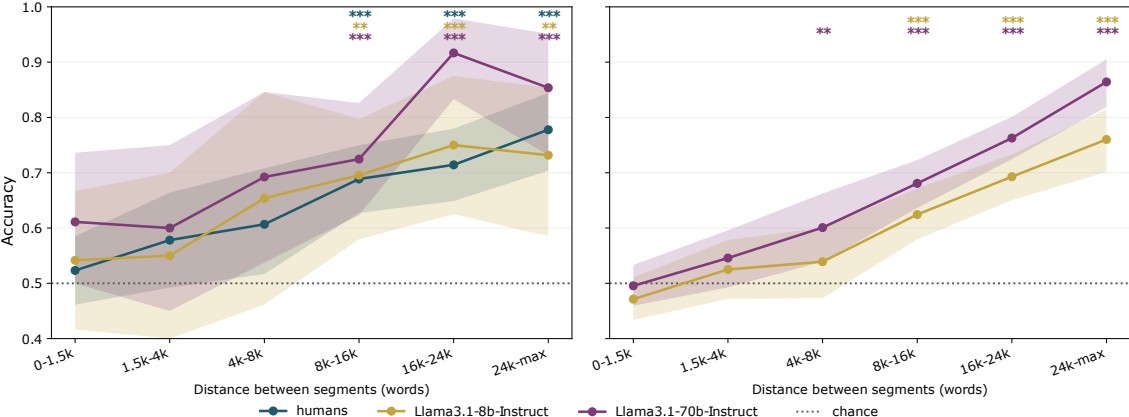

*Figure 2.* **Temporal order judgment accuracy vs. distance on *The Murder of Roger Ackroyd* (original vs. sentence-block-shuffled).** Distance-binned accuracy (mean ± 95% CI). **Left:** Humans, Llama 3.1-8B-Instruct, and Llama 3.1-70B-Instruct evaluated on the same 296 segment pairs from the original book. **Right:** Llama 3.1-8B-Instruct and Llama 3.1-70B-Instruct evaluated on four-sentence-block shuffled variants of *The Murder of Roger Ackroyd*, averaged across four random shuffles of blocks of four sentences. Asterisks indicate significance levels in a binomial test relative to chance performance in each bin.

To obtain temporal readout directions that are not specific to any particular narrative content, we learn head-specific directions on a *random-word* document (60k i.i.d. English nouns), and evaluate them on disjoint documents (e.g., a cleaned recent *House of Commons* transcript and sentence-shuffled book variants). For each $(\ell, h)$, we fit two readout-directions $w_{\text{ret}}^{(\ell,h)}$ and $w_{\text{enc}}^{(\ell,h)}$ as the temporally-oriented top principal component on the retrieval representations $R_{\text{ret,train}}^{(\ell,h)}$ and encoding representations $R_{\text{enc,train}}^{(\ell,h)}$ respectively from the training dataset. In Appendices A.3 and A.2, we discuss this choice and compare it with linear probing as an alternative to find $w_{\text{train},(\cdot)}^{(\ell,h)}$. Since eigenvectors are defined only up to a sign, we resolve the ambiguity by orienting $w$ for positive temporal alignment on the training set:

$$\tau_{\text{train}}^{(\ell,h)} = \hat{\tau}\left(R_{(\cdot),\text{train}}^{(\ell,h)};\ w^{(\ell,h)}\right), \qquad (7)$$

$$w_{\text{train}}^{(\ell,h)} \leftarrow \text{sign}\left(\tau_{\text{train}}^{(\ell,h)}\right) w^{(\ell,h)}. \qquad (8)$$

After this orientation step, a negative $\tau_{\text{K}}$ on held-out datasets reflects a genuine reversal relative to the fixed training orientation: a lack of generalization.

For each held-out dataset d with $N_{\text{d}}$ segments ordered by ground-truth text position, we compute temporal order scores $\tau_{\text{enc}}^{(\ell,h)}$ and $\tau_{\text{ret}}^{(\ell,h)}$:

$$\tau_{(\cdot)}^{(\ell,h)}(\text{d}) = \hat{\tau}\left(R_{(\cdot),\text{d}}^{(\ell,h)};\ w_{\text{train},(\cdot)}^{(\ell,h)}\right) \qquad (9)$$

and aggregate across held-out datasets $\text{d} \in \text{D}_{\text{test}}$ by averaging (reporting bootstrapped 95% CIs across datasets and/or segments).

We find high temporal information present in the encoding-phase value-representations (Figure 3) with maxima in layer

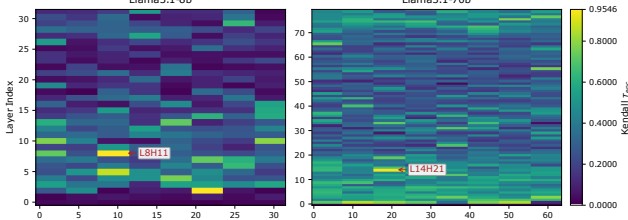

*Figure 3.* **Encoding-phase temporal information in the value cache of each attention head.** We use readout-directions fitted to the training-set to evaluate the rank-correlation $\tau$ of positional information between the random word training set and the five held-out test datasets. Data are averaged across all test data. Left: Llama 3.1-8B-Instruct, Right: Llama 3.1-70B-Instruct. Head indices are shown in query-head coordinates (where 4 query-heads share a KV-cache in Llama 3.1-8B-Instruct and 8 in Llama 3.1-70B-Instruct).

2 and 8 for Llama 3.1-8B-Instruct and layer 14 in Llama 3.1-70B-Instruct. In retrieval-phase representations, we find that a single query head stands out in both the 8b and the 70b models: L8H11 and L14H21 respectively (Figure 4).

An attention head can exhibit temporal structure in $R_{\text{ret}}^{(\ell,h)}$ for at least two qualitatively different reasons: (i) *primary* temporal information is retrieved by that head from its own stored value-cache formed during encoding (i.e., temporal structure is already present in $R_{\text{enc}}^{(\ell,h)}$ along direction $w_{\text{train}}^{(\ell,h)}$ and is simply reinstated at retrieval), or (ii) *secondary* temporal information can be inherited at retrieval from upstream computations (e.g., a previous layer writes temporally organized information into the residual stream at the retrieval segment span, which many subsequent heads can then inherit through the residual stream), without being present in that head's value-cache during the encoding phase. To

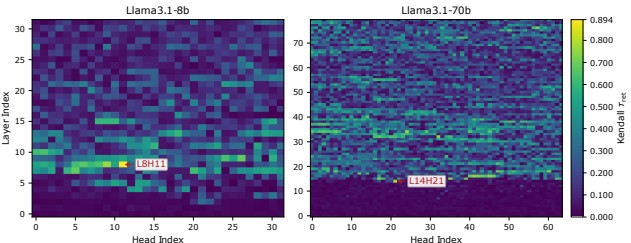

Figure 4. **Retrieval phase temporal information in each attention head.** We use readout-directions fitted to the training-set and evaluate rank-correlations, averaged across the five held-out test datasets. Left: Llama 3.1-8B-Instruct, Right: Llama 3.1-70B-Instruct

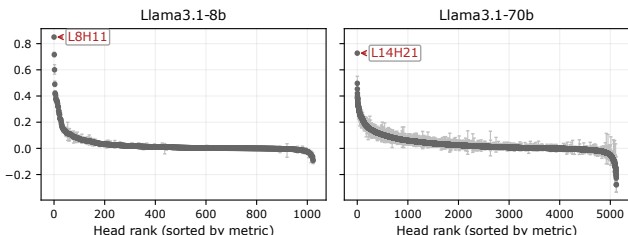

Figure 5. **Distribution of temporal reinstatement scores across all attention heads** in Llama 3.1-8B-Instruct and Llama 3.1-70B-Instruct respectively. See Figure 18 for an associated heatmap figure.

find heads with *primary* reinstated temporal information, it is required that a head's temporal readout direction $w_{\text{train}}^{(\ell,h)}$ generalizes to *both* retrieval and encoding representation matrices under the same fixed readout direction.

We define a head-level temporal reinstatement score $TRS^{(\ell,h)}$ that is large only when temporal alignment is high across both encoding and retrieval along $w_{\text{train,ret}}^{(\ell,h)}$ while penalizing the presence of sign flips:

$$\text{TRS}^{(\ell,h)} = \text{sign}\left(\min\left(\overline{\tau_{\text{enc}(w_{ret})}^{(\ell,h)}}, \overline{\tau_{\text{ret}}^{(\ell,h)}}\right)\right) \cdot \left|\overline{\tau_{\text{enc}(w_{ret})}^{(\ell,h)}} \, \overline{\tau_{\text{ret}}^{(\ell,h)}}\right| \tag{10}$$

where $\overline{\tau_{\text{enc}}^{(\ell,h)}}$ and $\overline{\tau_{\text{ret}}^{(\ell,h)}}$ denote averages across $d \in \mathcal{D}_{\text{test}}$. The sign term $\text{sign}(\min(\cdot))$ ensures that in the presence of a lack of generalization of the sign of $w^{(\ell,h)}$ to the test set for either the encoding or retrieval representations, we assign a negative time-reinstatement score. We confirm that identified time-reinstatement heads are indeed doing retrieval in Appendix A.6.

**A single head dominates in the reinstatement of temporal order information.** For each model, we find two remarkable outliers, suggesting that time-reinstatement may be localized in a single attention head. For the 8b model, the highest TRS is found in L8H11; for the 70b model, the highest TRS is in L14H21, with large margins to the next heads (see Figure 5 and Appendix A.7). An analysis of attention distribution at these heads further confirms they are acting as retrieval heads, with an average of 64% of attention mass placed on the encoding-span of segments for the 8b model and 39% for the 70b model.

**Reinstated temporal information does not depend on RoPE.** To rule out that the localized temporal code is merely a consequence of the model's rotary positional encoding (RoPE), we repeat the localization analysis in Llama 3.1-8B-Instruct with RoPE disabled, using randomly sampled token sequences (see Appendix A.1). The temporal information carried by the time-reinstatement head is preserved (even *strengthened*) when RoPE is removed, and the

PC1 direction in the top temporal reinstatement head under these conditions still aligns with $\hat{w}$, the PC1 direction used in our main experiments (cosine similarity 0.69 between $w_{\text{NoPE}}^*$ and $\hat{w}$). The substantial remaining alignment after both the removal of RoPE and the introduction of a disjoint, content-free stimulus indicates that the temporal axis is tied neither to the positional scaffold nor to any particular input distribution. A randomly initialized model, by contrast, shows little temporal structure with or without RoPE, indicating that this structure is likely acquired through training. Taken together, these results indicate that the temporal code is *learned*, not inherited from RoPE. We do not evaluate SORT behavior without RoPE, since removing it substantially shifts the model's output distribution, which would render behavioral comparisons uninterpretable rather than informative about temporal reinstatement.

### 5.3. Causal importance of temporal reinstatement heads

So far we have found (for each model) a special head and its principal temporal-coding direction that recovers temporal order in both its memory representations (value-cache) and its retrieval-phase representations, suggesting these heads are mostly reinstating temporal information. We verified in Appendix A.6 that these heads are indeed among the top retrieval heads in the models. We now want to show that the heads are uniquely causally implicated in the model's task performance. To do so, we perform causal interventions that remove or amplify the time-coding directions in the retrieval-phase representations of each head.

We test whether temporal ordering behavior depends *causally* on the temporal readout direction identified in the time-reinstatement heads. We denote the head with maximal TRS in a given model by $(\ell^\star, h^\star)$, and define its temporal-readout direction as $\hat{w} \equiv w_{\text{train,pc1}}^{(\ell^\star,h^\star)} \in \mathbb{R}^{d_h}$.

We intervene by (a) removing variance of the head's output along $\hat{w}$ *only* at the retrieval-phase occurrences of the queried segments (both A and B), leaving other token positions unaffected, and (b) instead of removing variance, we scale it by a factor $\alpha$.

Let $\mathcal{T}_A^{\mathrm{ret}}$ and $\mathcal{T}_B^{\mathrm{ret}}$ denote the token index sets corresponding to the retrieval-phase span of segments A and B, respectively, and let

$$\mathcal{T}_{\mathrm{seg}}^{\mathrm{ret}} = \mathcal{T}_A^{\mathrm{ret}} \cup \mathcal{T}_B^{\mathrm{ret}}. \tag{11}$$

We define an intervention operator $\Pi_{\perp \hat{w}}$ that removes the component along $\hat{w}$:

$$\Pi_{\perp \hat{w}}(x) = x - \langle x, \hat{w} \rangle \hat{w} = \left( I - \hat{w}\hat{w}^{\top} \right) x. \tag{12}$$

The intervened head outputs $\tilde{o}_t$ are then

$$\tilde{o}_t = \begin{cases} \Pi_{\perp \hat{w}}(o_t), & t \in \mathcal{T}_{\mathrm{seg}}^{\mathrm{ret}}, \\ o_t, & \text{otherwise.} \end{cases} \tag{13}$$

Eq. (13) is applied online during the forward pass only at layer $\ell^{\star}$ for head $h^{\star}$. Because the intervention is restricted to the retrieval spans of the segments, it targets reinstatement at test time, while preserving the model's processing of the long encoding context.

In a separate experiment, we scale the temporal component along $\hat{w}$ (and random $w$ as a control) by a range of factors $\alpha \in \mathbb{R}$ while leaving the orthogonal subspace unchanged:

$$\tilde{o}_t = \begin{cases} o_t + (\alpha - 1)\langle o_t, w \rangle w, & t \in \mathcal{T}_{\mathrm{seg}}^{\mathrm{ret}}, \\ o_t, & \text{otherwise.} \end{cases} \tag{14}$$

**Directional removal reduces average performance to near-chance.** Projecting out the time-reinstatement direction reduces performance in both models by 10-15%. Differences are significant under a McNemar test against the unablated task performance. In Llama 3.1-8B-Instruct, an analysis of all attention heads reveals that the top retrieval head is the only one that is causally necessary for high accuracy on the task (see Figure 6). For Llama 3.1-70B-Instruct, we show this based on a subset of 180 attention heads, consisting of the top 20 TRS heads and 160 randomly sampled heads. Notably, in Llama 3.1-70B-Instruct, ablating other heads' temporal principal directions *improves* performance, which indicates greater interference in the model.

**Amplification can further improve ordering task performance.** Fig. 7 shows that scaling $\alpha$ can improve accuracy. In Llama 3.1-8B-Instruct, we find that increasing gain on L8H11 by a factor of 1.75 increases accuracy by up to 9%.

### 5.4. Generalization to other models and ordering tasks

In two additional smaller-scale experiments, we find preliminary evidence that the mechanism is not specific to one model or task format. An outlier time-reinstatement head, with a clear margin to the next head and a causal effect on ordering accuracy, is also present in Mistral-7B-v0.2 and Qwen2.5-7B (Appendix A.8). We further find that in

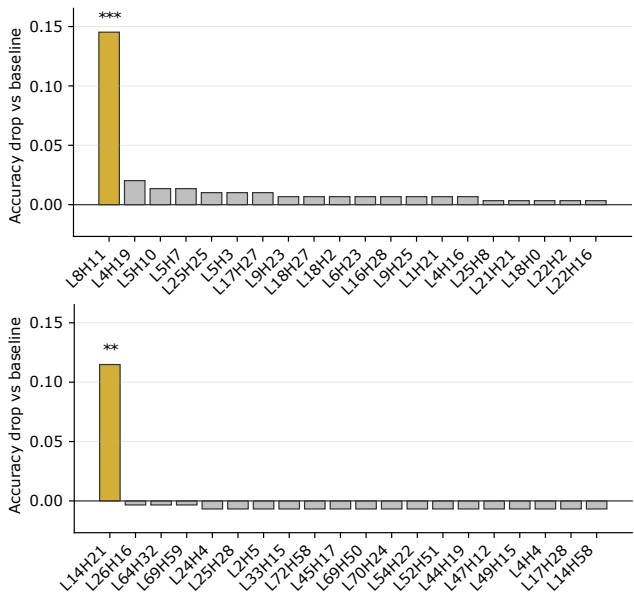

*Figure 6.* Reduction in accuracy after projecting out attention heads' principal temporal direction, showing (in order) the heads with the 20 highest reductions in accuracy. The top corresponds to the 8b model (all heads), bottom shows the 70b model (top 20 TRS heads and 160 randomly sampled heads were included in this analysis). Asterisks indicate level of statistical significance in a pairwise McNemar test against the baseline (no ablation). *** corresponds to $p < 0.001$ in a McNemar-test, after Benjamini-Hochberg correction for multiple tests.

Llama 3.1-8B-Instruct, the causal implication of L8H11 is not specific to the binary SORT task, but extends to a four-segment ordering task (Appendix A.9).

## 6. Discussion

We found that across both models, temporal order judgments rely on a highly localized mechanism: a single attention head that stores a low-dimensional temporal code during encoding reinstates it during retrieval. Causal interventions that remove or amplify this direction during the retrieval phase substantially modulate accuracy, indicating that *temporal* context reinstatement is not merely correlated with performance but in LLMs, it is causally important for temporal order memory.

Temporal ordering accuracy remains largely unchanged under sentence-block shuffling, suggesting that narrative coherence and causal schemas are not the primary drivers of model performance in our setup. This does not rule out semantic or schema-based strategies in other prompts or tasks, but it shows that an explicit temporal trace plus reinstatement can be sufficient for robust long-context order judgments.

The identified temporal axes provide a concrete target for comparison with biological accounts of tempo-

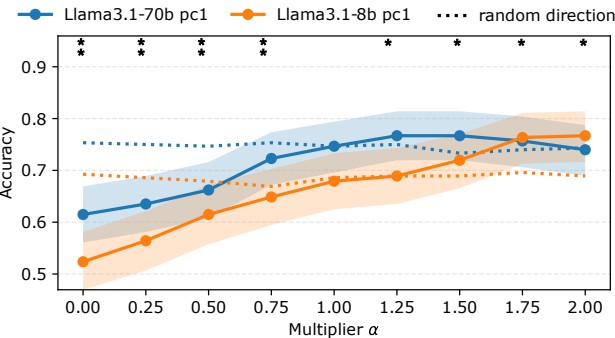

*Figure 7.* Effect of increasing/decreasing gain on PC1 in the top time-reinstatement head (solid line) compared with randomly chosen directions. Shaded areas correspond to bootstrapped 95% confidence intervals. Asterisks show statistical significance of $p < 0.05$ in paired one-sided McNemar tests with the baseline ($\alpha = 1.0$) after BH-correction (Benjamini & Hochberg, 1995).

ral representation, including time cells, ramping codes, and low-dimensional representational drift in hippocampal–entorhinal circuits (Umbach et al., 2020; Rolls & Mills, 2019; Wang et al., 2025). More broadly, our analyses demonstrate that long-context mechanistic interpretability can localize temporal "episodic-like" circuitry in transformer models and support causal tests of hypothesized retrieval mechanisms.

An important open question is how these temporal directions arise and when they are recruited for more cognitively complex tasks beyond order judgments—e.g., in event segmentation, schema induction, or causal reasoning over narratives—and how their influence varies with context length and interference. Answering these questions could yield insights for both long-context behavior in models and temporal episodic memory in humans. Additionally, better understanding and improving the sample-by-sample agreement between humans and models is an important direction for future work that aims to develop improved cognitive models of episodic memory. One concrete step in this direction is to minimize the mismatch in evaluation settings between humans and models by letting models judge multiple segment pairs within the same context.

*Limitations.* We focus on two long-context Llama-3 models and one canonical temporal-order memory task. This controlled setting clarifies size-related differences. While we present suggestive evidence of temporal context reinstatement in other model families and beyond binary order judgments, it remains unclear whether localized temporal reinstatement occurs in hybrid models, and how temporal reinstatement affects other tasks. Other behaviors may recruit (or bypass) similar dynamics; extending our methodology to additional tasks is a clear next step for mapping when temporal reinstatement supports behavior and for linking mechanism to normative accounts of *why* temporal reinstate-

ment heads emerge.

**A note on possible data contamination.** Since the LLMs we test were released after the book became part of the public domain, the LLMs may have been trained on the book text which may affect the task accuracy we report. To alleviate this concern, we performed additional experiments on several shuffled versions of the book which are unlikely to have been used for LLM training. The shuffled texts show similar accuracy to that of the original book text, ruling out the possibility that possible data contamination substantially affects our findings.

## Impact Statement

Further clarifying the connections between episodic memory mechanisms and in-context memory mechanisms in LLMs has the potential to inform a new class of computational models of memory, as well as new memory architectures for long-context generalization (see Fountas et al. (2025)). Our work investigates one important aspect of episodic memory in LLMs' in-context memory behaviorally, representationally, and mechanistically. We also show that long-context mechanistic interpretability experiments are feasible in the context of memory experiments. Our code is available at https://github.com/mathispink/temporal-context-reinstatement. To enable more long-context memory research of this kind, we additionally release a flexible toolkit: ICMemory-Kit (https://github.com/mathispink/icmemory-kit).

## Acknowledgements

This work is funded in part by the Deutsche Forschungsgemeinschaft (DFG, German Re- search Foundation) – GRK 2853/1 "Neuroexplicit Models of Language, Vision, and Action" - project number 471607914 and the C.V. Starr Fellowship at Princeton University.

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

## A. Appendix

### A.1. Temporal information does not depend on RoPE

We analyze temporal order information in Llama3.1-8b-Instruct on 10 different 94k random token sequences without using a chat template, **with and without RoPE enabled** (see Figure 8). Following our PCA-based methodology, we compute the time-oriented PC1 readout direction in the value-cache for each head on the first random sequence, and then validate it against the remaining 9 random sequences, averaging the rank correlations. We find that when RoPE is enabled, *less* temporal information is encoded in the memory representations. We compare the PC1 direction for the time-reinstatement head L8H11 and find for both RoPE enabled and disabled a cosine similarity of $0.69$ with the PC1-direction that we used in the experiments reported in this paper. This strongly suggests that the encoding of temporal/positional information does not depend on explicit positional encoding and is surprisingly stable beyond the fixed prompt templates used in our experiments.

In addition to this, we ran the same experiment on a randomly initialized version of the model without RoPE to see how much temporal information is given in a random architecture, the results of which can be seen in Figure 9. We find that little temporal information is encoded in the attention heads' value cache without any training, suggesting that positional scaffolding may help during training but is not the primary source of positional/temporal information after training. We find that in untrained models, RoPE leads to slightly higher temporal information. Together, these findings mirror and extend recent findings (Gelberg et al., 2025).

### A.2. PCA and linear probing of position

We find that our time-reinstatement score results do not change substantially when relying on linear probing of segment position instead of using PC1. We use a large ridge regularization factor of $1e5$ and use $R_{\text{enc}}$ and $Rret$ to predict the positional start index associated with each row. Like in our main experiments, we use the random-words sequence as training data and present generalization results averaged across 5 held-out documents.

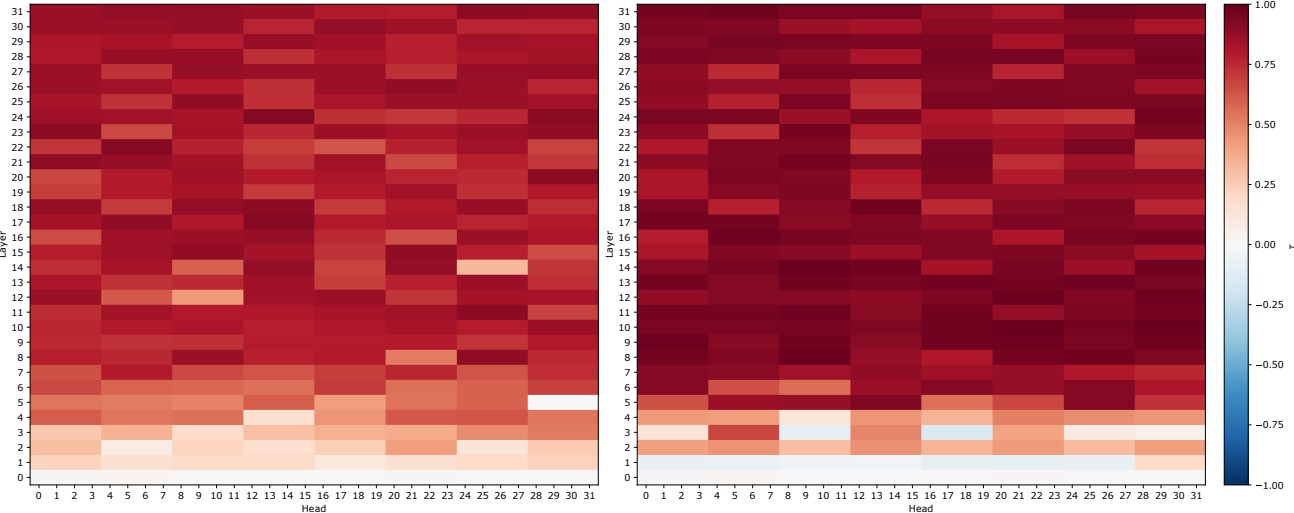

*Figure 8.* Llama 3.1-8B-Instruct with RoPE enabled (left) and RoPE disabled (right), evaluated on 10 random token sequences of 94k tokens each. Shown is the temporal order information (Kendall's $\tau$) along PC1 trained on the first random sequence and evaluated on the remaining 9 sequences. We see *more* not less temporal information without RoPE, indicating that the temporal information that is reinstated in the time-reinstatement heads is probably not related to any explicit positional encoding.

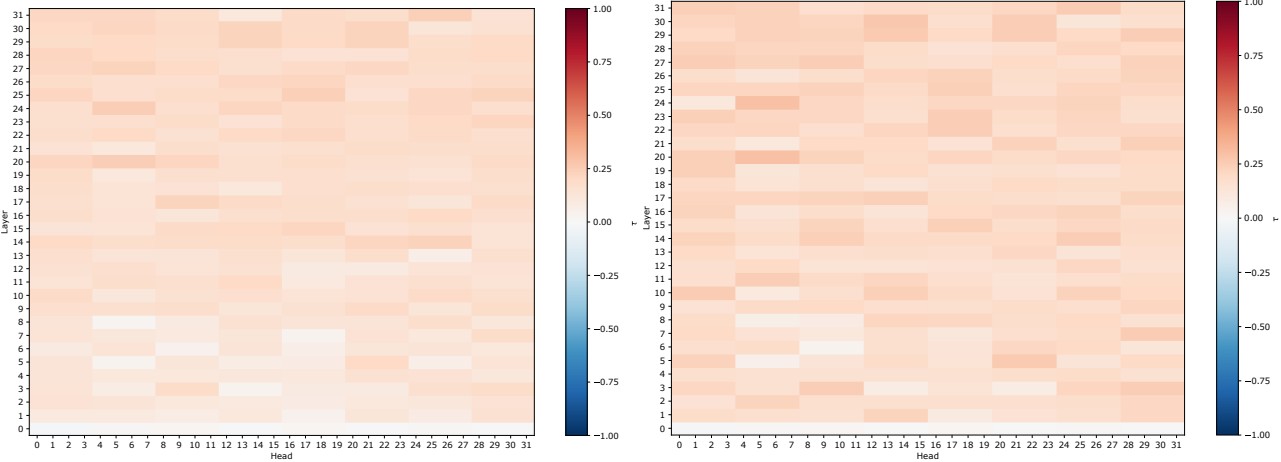

*Figure 9.* Same setup as in Figure 8, but for a randomly initialized Llama3-8b model. Left: *without RoPE*. Right: *with RoPE*.

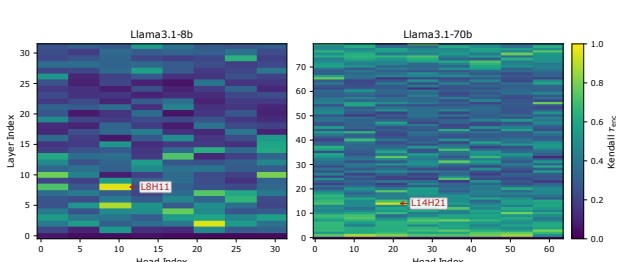

*Figure 10.* **Encoding-phase temporal information.** Equivalent figure to 3 (temporal information in encoding-phase representations) but instead of PC1, we use ridge-regression linear probing of segment position.

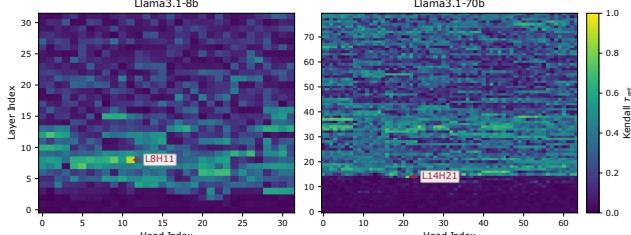

*Figure 11.* **Retrieval-phase temporal information.** Equivalent figure to 4 (temporal information in retrieval-phase representations) but instead of PC1, we use ridge-regression linear probing of segment position.

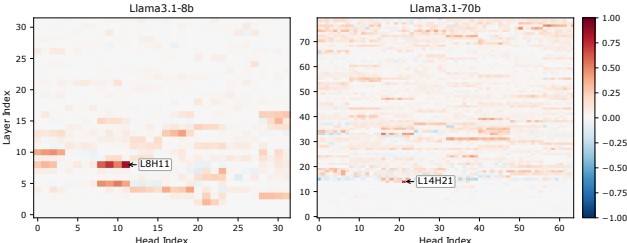

*Figure 12.* Distribution of temporal reinstatement scores across the two model, but instead of PC1, we use ridge-regression linear probing of segment position.

These results suggest that PC1 can be used in place of a ridge regression with a high regularization strength to find the same localization of temporal reinstatement heads.

### A.3. PCA: Relation to spectral seriation

We can motivate why PCA recovers temporal order information by framing the task as a seriation problem - a class of problems initially described in the context of temporally dating a collection of archeological items based on their feature similarities (Robinson, 1951). We evaluate temporal order memory representations via a spectral seriation approach (Fogel et al., 2016). Framing it in this way closely relates it to the analyses of (Park et al., 2025) on in-context learning which showed that representational graph structure encodes the LLM prompts' underlying data generating processes. (Park et al., 2025) show that principal components become spectral readout directions if the model is minimizing dirichlet energy, which suggests that our PCA based ordering is a special form of spectral seriation under additional assumptions.

A *seriation* problem asks: given pairwise similarities between items, recover (up to reversal) an ordering consistent with an underlying 1D latent axis (e.g. temporal order, or ranking). In our setting, if a heads' representations imply a temporal code, then the similarity structure of the rows of $R_{\text{enc}}^{(\ell,h)}$ and/or $R_{\text{ret}}^{(\ell,h)}$ should admit a 1D ordering aligned with the ground-truth indices $1, 2, \ldots, N$.

#### A.3.1. SPECTRAL SERIATION BASED TEMPORAL ORDER JUDGMENT

Here we describe how spectral seriation implies an ordering. Together with findings from (Park et al., 2025), this could point to an explanation of why PC1 often recovers a correct ordering.

For a given representation matrix $R \in \mathbb{R}^{N \times d}$ (standing for either $R_{\text{enc}}^{(\ell,h)}$ or $R_{\text{ret}}^{(\ell,h)}$), we can compute a cosine-similarity representational similarity matrix (RSM)

$$S_{ij} = \frac{\frac{\langle R_{i,:}, R_{j,:}\rangle}{\|R_{i,:}\|\,\|R_{j,:}\|} + 1}{2} \in [0,1], \quad (15)$$

Interpreting $S$ as a weighted adjacency matrix of an implied representational graph, we form the degree matrix $D = \text{diag}(S\mathbf{1})$ and the unnormalized graph Laplacian

$$L = D - S. \quad (16)$$

$f \in \mathbb{R}^N$ is the *Fiedler vector*, i.e., the eigenvector associated with the smallest non-trivial eigenvalue of $L$. Spectral seriation induces an ordering by sorting the entries of $f$:

$$\hat{\pi} = \text{argsort}(f) \in S_N. \quad (17)$$

Since in this case, we know the ground-truth ordering, we simply evaluate alignment to the ground-truth temporal order by Kendall's rank correlation ($\tau$) between the induced sorting-order and the identity permutation:

$$\tau_{\text{K,spectral}}(R) = |\tau_{\text{K}}(\pi(f), (1, 2, \ldots, N))|. \quad (18)$$

We take the absolute value because eigenvectors are defined up to a sign, thus requiring soft supervision in spectral seriation to orient the vector's sign for correct ordering.

### A.4. Generalization of random-word sequence PC1 (input independence of attention heads)

Here, we show the generalization of temporal order along the PC1 fitted to the random word dataset.

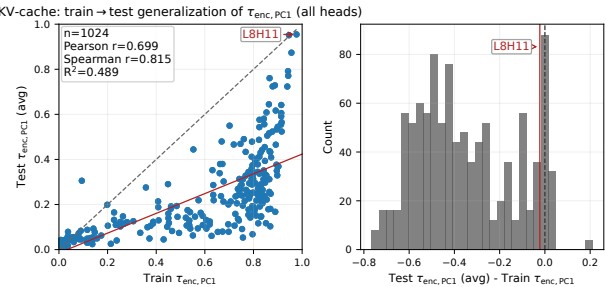

*Figure 13.* Llama 3.1-8B-Instruct: Generalization of temporal order readout directions in the value cache (encoding-phase) from random-word sequence fitted PC1 to that direction readout across the five test documents.

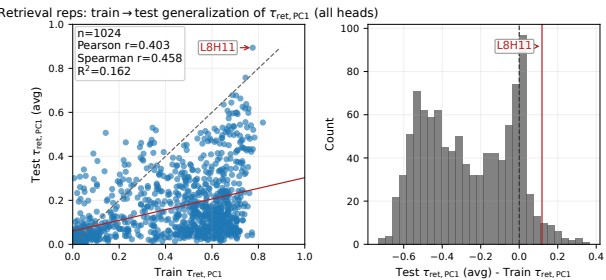

*Figure 14.* Llama 3.1-8B-Instruct: Generalization of temporal order readout directions in the retrieval-phase representations from random-word sequence fitted PC1 to that direction readout across the five test documents.

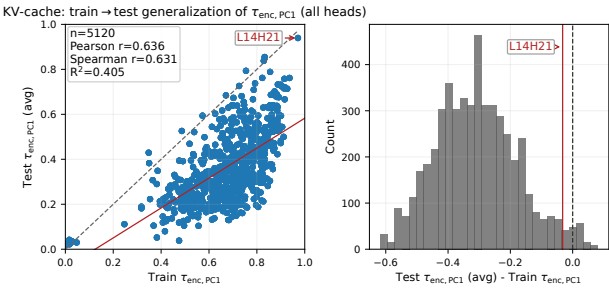

*Figure 15.* Llama 3.1-70B-Instruct: Generalization of temporal order readout directions in the value cache (encoding-phase) from random-word sequence fitted PC1 to that direction readout across the five test documents.

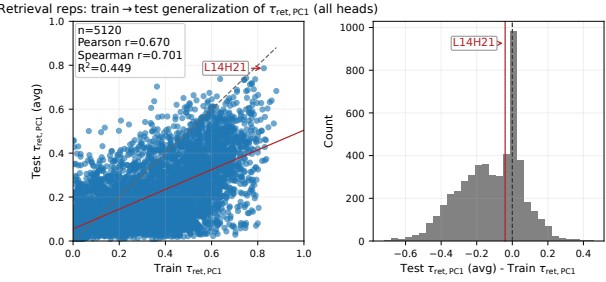

*Figure 16.* Llama 3.1-70B-Instruct: Generalization of temporal order readout directions in the retrieval-phase representations from random-word sequence fitted PC1 to that direction readout across the five test documents.

### A.5. Dataset details

***The Murder of Roger Ackroyd* (original order; human+model).** Our primary dataset is based on *The Murder of Roger Ackroyd* (approximately 60k words). We construct 296 query pairs of 50-word segments from the book. These 296 pairs are judged by human participants and are also evaluated by the models, enabling direct comparison of distance-dependent temporal order accuracy between humans and LLMs. Segment pairs span a range of distances $\Delta$ (in words) to probe the characteristic distance effect in temporal order memory (higher accuracy for widely separated segments and near-chance performance at small separations).

***The Murder of Roger Ackroyd* sentence-shuffled variants (models only).** To break narrative and causal structure while preserving lexical content, we create four sentence-level shuffled versions of *The Murder of Roger Ackroyd* (seeds 1–4). Each variant is produced by permuting the order of blocks of four sentences in the original text, keeping local sentence-block-internal word order intact but breaking up narrative structure and causal chains. For each shuffled variant, we generate a corresponding set of 600 segment pairs of 50-word segments sampled from that shuffled text

and evaluate only the models on these pairs. This control tests whether model performance critically depends on narrative coherence or causal schemas: if order judgments in models relies on narrative or causal reasoning, shuffling should impair performance. We additionally use this dataset to validate readouts of temporal-information fitted to a random word sequence.

**House of Commons transcript (models only).** To test whether the mechanistic signatures of temporal order retrieval generalize beyond the literary text and its shuffled variants, we additionally use a long House of Commons transcript from 2025-09-08, truncated to 100k tokens. We cleaned any enumerations, long lists of names and other explicit temporal or positional markers from this text. From the cleaned document, we sample 600 query pairs of 50-word segments and evaluate the models on the same temporal order judgment task as in the book setting.

**60k random-word sequence (mechanistic "training" dataset).** To obtain head-specific temporal readout directions independent of book content, we construct a 60k-word i.i.d. sequence of English nouns. We partition the sequence into contiguous and non-overlapping 50-word segments, yielding 1200 segments, and use the resulting segment representations during encoding and retrieval as a training dataset for fitting linear readouts (e.g., PCA-based temporal directions). These readout directions are then held fixed and evaluated on the other documents (e.g., House of Commons and shuffled *The Murder of Roger Ackroyd*) to quantify cross-dataset generalization (narrative-independence) of temporal structure in representations of the models.

### A.6. Attention-scores retrieval analysis

To confirm that TRS heads are doing retrieval, we look at whether their attention scores fall within the span of the segments in the encoding phase. We look at the average attention mass fraction falling into the encoding-span of the segments.

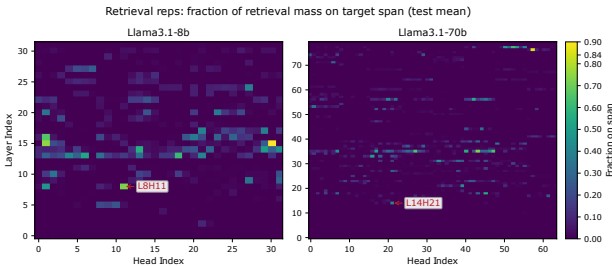

*Figure 17.* Retrieval-phase attention mass fraction falling within the span of the segments, averaged over test datasets. Confirming that L8H11 in Llama 3.1-8B-Instruct and L14H21 in Llama 3.1-70B-Instruct are among the top retrieval heads.

## A.7. Temporal Reinstatement Score Heatmaps

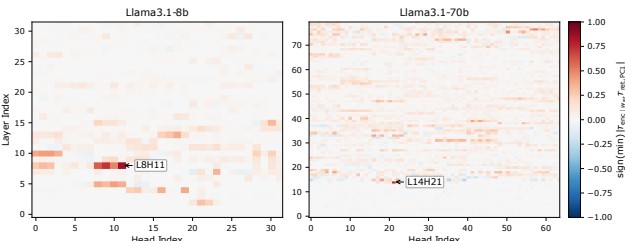

*Figure 18.* Distribution of temporal reinstatement scores across the two model.

## A.8. Cross-model temporal reinstatement and ablation

To test whether single-head temporal reinstatement is specific to the Llama-3 family, we repeat the localization and parts of the causal analyses on two 7B-parameter models from different families and training pipelines: Mistral-7B-v0.2 and Qwen2.5-7B. Additionally, since the Mistral model has a maximum context length of 32k, we use 8k tokens context rather than the 94k+ context used in the main experiments. We fit head-specific temporally aligned PC1 directions on a shortened i.i.d. random-word sequence (Appendix A.5), compute the per-head temporal reinstatement score $\text{TRS}^{(\ell,h)}$ exactly as in Eq. (10), and we remove the top reinstatement head's contribution to the residual stream during the retrieval-phase spans of the queried segments. Because both the encoding context and the segment-distance range are smaller than in our main long-context experiments, the absolute TRS values and ablation effect sizes reported here are not directly comparable to those in the main text for the Llama3 models and should be read only as a preliminary qualitative replication.

In both models a single head again dominates the temporal reinstatement score, with a clear margin to the next-ranked head. For Mistral-7B-v0.2 the maximum is at L7H18 with $\text{TRS} = 0.53$ and a gap of $0.22$ to the second-ranked head; for Qwen2.5-7B it is at L14H0 with $\text{TRS} = 0.40$ and a gap of $0.13$. Removing these heads at the retrieval spans reduces SORT accuracy from $0.60$ to $0.50$ for Mistral-7B-v0.2 and from $0.65$ to $0.60$ for Qwen2.5-7B. These drops are smaller in absolute terms than in the long-context Llama experiments, consistent with the much shorter context and reduced distance range, but the qualitative signature, a single outlier reinstatement head whose removal degrades ordering accuracy, is present in both independently trained model families.

## A.9. Multi-segment ordering

To test whether temporal reinstatement extends beyond the binary SORT format, we generalize the task from two segments to four. Under a random label assignment A-D, we present four 50-word segments. Instead of a single binary comparison, the model must produce the full ordering: we score each of the $4! = 24$ candidate orderings by the log-probability the model assigns to the corresponding answer and take the argmax as the predicted ordering. Performance is quantified by the rank correlation (Kendall's $\tau$) between the predicted and ground-truth orderings, so that chance corresponds to $\tau \approx 0$ and a perfect ordering to $\tau = 1$.

In a preliminary experiment, we evaluate Llama 3.1-8B-Instruct on multi-segment ordering from two excerpt sizes from *The Picture of Dorian Gray*. At a short 500-word context the model orders the four segments well above chance ($\tau = 0.63$); at a 25k-word context, baseline performance is lower ($\tau = 0.31$). Removing the time-reinstatement head L8H11 during the retrieval phase reduces the ordering accuracy to $\tau = 0.54$ (95% bootstrap CI $[0.51, 0.56]$) at the 500-word context and to $\tau = 0.11$ at 25k words.

The asymmetry may suggest that single-head temporal reinstatement is the dominant mechanism for larger context lengths and segment distances, whereas short-context ordering may additionally recruit more distributed circuitry. We regard these results as preliminary: they cover a single model and one multi-segment configuration, and a fuller characterization across models, context lengths, and segment counts is left to future work.

## A.10. Human study details

**Participant compensation.** Participants were compensated via a lottery system with a chance to win a gift card to a popular book store. The expected value of the compensation came out to $12 per hour.

**Study design.** Each participant completed an online survey. First, the participant consented to the study, read a brief set of instructions, and completed a brief survey, including a question regarding when the participant finished reading the book. The complete set of survey questions is listed below. Each participant was then asked to answer "Which segment occurred first in the book?" for about 10 randomly chosen text segment pairs from a total set of 540 unique segment pairs sampled from the whole book. We chose to present a sample number of trials to each participant to minimize interference effects from repeated memory retrieval (Kliegl & Bäuml, 2021). The presentation order of the text segments was randomized across participants. In the end, each participant was asked 4 simple questions about the book plot to verify that the participant had indeed read the book. Each participant was only allowed to participate in the study once.

**Demographics questions.** The human participants were asked the following set of demographics questions before

beginning the experiment:

1. I have finished the book The Murder of Roger Ackroyd [Options: True/False]

2. On what date did you finish the book? [Calendar question type]

3. Did you read or listen to the book? [Options: Read/Listen]

4. Was this your first time reading / listening to the book? [Options: Yes / No]

5. What is your age? [Options: 18-25, 25-35, 35-45, 45-55, 55-65, 65+]

6. What gender do you identify with? [Options: Female/Male/Other]

7. What is your experience with the English language? [Options: Native / Fluent / Advanced / Intermediate / Beginner]

8. How many books did you read or listen to in the past year? [Options: 1-2 / 3-5 / 6-10 / 10+]

We use the responses above to determine the number of days that have passed since finishing the book, and make this information available in the human dataset together with the responses.

**Inclusion criteria.** We include data from participants who answered at least 3 of 4 plot questions correctly, and finished reading the book within 30 days of participating in the study. These inclusion criteria result in 97 participants.

### A.11. Prompts used for model evaluation

We use different prompts depending on the type of document (the novel and its shuffled variants, the House of Commons transcript, and the random words sequence).

For the novel, the encoding-phase part of the prompt is given by:

> I need you to thoroughly read and comprehend the book The Murder of Roger Ackroyd. Here's the complete book: *complete book text*

Followed by the retrieval-phase part of the prompt:

> Your task is to recall which text segment, either A or B, appeared first in the book The Murder of Roger Ackroyd. Please read the segments carefully to remember in which order they appeared in

The Murder of Roger Ackroyd and respond with either A or B: Segment A: *segment A* Segment B: *segment B* Which of these, A or B, was first in the book The Murder of Roger Ackroyd?

For the documents unrelated to the novel, we use this encoding-phase prompt:

> I need you to thoroughly read and comprehend the following document. Here's the text: *document text*

Followed by this retrieval-phase prompt:

> Your task is to recall which of two previously seen text segments, either A or B, appeared first. Please read the segments carefully to remember in which order they appeared in the document and respond with either A or B. Segment A: *segment A* Segment B: *segment B* Which of these, A or B, was first in the document?

While current analyses used English proficiency and days-since-finishing, the remaining demographic items were collected for future stratified analyses. Audiobook listeners were not excluded in the analyses.

