# OpenReview forum: "Temporal Context Reinstatement Drives Episodic-Like Order Memory in Long-Context Language Models"
_ICML.cc/2026/Conference — ICML 2026 regular_

### Official Review · Reviewer_wUZw · 2026-03-11

**Soundness:** 2
**Presentation:** 3
**Significance:** 3
**Originality:** 3
**Overall Recommendation:** 5
**Confidence:** 4

**Summary:**

The paper investigates how LLMs can tell apart whether sentence A comes before sentence B in long sequences of text. Specifically, this type of testing is done over a full-length novel, and the responses are compared to those of human participants who have read the book. The authors report qualitative similarities between how humans and LLMs perform on this task. Through a series of analyses, the authors show that an individual attention head is responsible for reinstating the temporal context, allowing the model to successfully do this task.

**Compliance With Llm Reviewing Policy:**

Affirmed.

**Final Justification:**

The authors have made important clarifications during the rebuttal. The paper was already in good shape during the initial submission, with exciting findings. However, the justification for the framing has improved considerably since, which is why I updated my score.

**Key Questions For Authors:**

The problem that is studied is not only relevant for long-contexts but for any sequence really. Do you think the identified circuitry operates only on very long contexts? Or can we have a setup with e.g. 10 sentences and find that the same attention heads are responsible for recruiting the right answers. I think either result would be highly interesting and informative.
- I do not fully understand Figure 3. There are only 8 columns for Llama 8B. However, you denote 32 heads on the x-axis. A similar mismatch exists in this figure for the 70B model.
- Do you always provide LLMs with the book followed by a single question, or do they also receive the sequence of questions in context like human participants do? The choice here has implications for interpreting similarities.
 - I really enjoyed reading the localisation experiments. However, I was a bit confused with the paragraph under Figure 4. Can the secondary temporal information also not encode both retrieval and encoding extractions.
- Why does the score not go as high in  Figure 5 for the larger model?
- The demographic questions ask a lot more than whether and when participants have finished the book. Why were the other questions asked, and have you used the rest of the information in any way? Was the data from the audio book listeners included?

**Limitations:**

I outlined most limitations I saw in the Weaknesses section above, which are mostly about the framing.

**Strengths And Weaknesses:**

**Strengths**
- Relevant literature is discussed under Links between LLMs and episodic memory.
- The collected human dataset is a very interesting one, and I believe it will be of use to multiple communities.
- There are several elegant manipulations/controls throughout the paper. Shuffling experiments that maintain the local structure, and the random word corpus used to fit the probes are two examples. Disabling ROPE in Appendix A.1. was similarly informative and elegant.
- To my knowledge, these types of "temporal retrieval" heads have yet not been identified by the community. If the findings are robust, they would be very interesting to the mechanistic interpretability community. In particular, that there is only a *single* head in each model that carries out this function appears to be a unique property, compared to most other attention heads (e.g. typically there are several induction heads/ function vector heads etc.)

**Weaknesses**
- The framing makes several large assumptions and leads to a disconnect between what is studied and how the findings are being interpreted, specifically about human episodic memory and the current operationalisation for LLMs. I provide specific points below. ***I would like to note that I don't think such strong language suggesting LLMs as models for human episodic memory is necessary at all for this paper. The paper offers interesting findings in its own right. Therefore, I encourage the authors to pose the human vs LLM analyses merely as a qualitative comparison and drop the currently unbacked claims of LLMs serving as models for human episodic memory.***
	- There appears to be a goal to use LLMs as mechanistic models for how human episodic memory works. This comparison fails at two different levels. First, by design, we know that the differences of how transformers and humans process sequences differ in significant ways that has specific implications for episodic memory. The transformer always has  *full access* to the entire sequence from the past, which is not the case for humans (e.g. I simply have no memory of some lunch I had on a particular day some months ago). This poses non-overlapping challenges for the two systems to solve episodic tasks. In addition, while I agree that the studied task itself has an episodic nature, episodic memory research encompasses many other core settings as well.
	- The qualitative comparison between human and LLM performance is very interesting! But I don't see it as a main analysis that serves to test LLMs as a model of human episodic memory. While this could serve as a posterior predictive check, I would like to see when you condition the LLM and the humans on the same input and the question, what is the likelihood that the LLM generates the human response? If you have some questions that repeat across different humans, you can also calculate a noise ceiling of predictivity and show your model fits with respect to that.
	- If one of the goals is to test LLMs as models of human episodic memory, I believe you should also compare your model against other prominent ones from the literature. Even if you don't go ahead with these comparisons, I encourage the authors to familiarise themselves more with the literature on models of the human memory system.
- I am unsure about the statistical soundness of the intervention results. Specifically those that are presented in Figure 7. Are there any statistical tests here? PC interventions do not seem significantly different.

**If the authors can address the two points I have raised above, I would be willing to increase my score.**

---

> ### Author Rebuttal · Authors · 2026-03-31
>
> Thank you for the careful reading and for the constructive critique. We especially appreciate your positive comments on the human dataset, the shuffling controls, the random-word probe fitting, the RoPE analysis, and the novelty of the temporal reinstatement head findings. We also agree that the framing should be more careful, and will revise the introduction/discussion accordingly.
>
> ### **Framing**
> Our work is motivated by prior work (Gershman et al. 2025; Ji-An et al. 2024; Park et al. 2025; Kolling et al. 2025), and our goal is to provide deeper insights into mechanisms of reinstatement in an episodic memory-related task. We will clarify that we do not aim to present a full-blown cognitive model of human behavior. Relatedly, we believe that comparing to other cognitive models is out of scope for this work. Additionally, our understanding is that no existing cognitive model can process such a long narrative and perform recency judgments. If you are aware of works that we have missed in this space, we’re highly interested.
>
> To make our framing and motivation more critical, we will add that transformers retain lossless access to the past representations, whereas human episodic retrieval is selective, lossy, and subject to many additional constraints. Despite these dissimilarities, recent work (cited in the paper, as well as above) has increasingly studied in-context memory through an episodic-memory lens.
>
> ### **Statistical soundness of the scaling interventions.**
> We validated the intervention effects against the baseline using paired one-sided McNemar tests with Benjamini-Hochberg correction. Across both models, downscaling the PC1 component significantly degrades performance, whereas random-direction controls produce no significant effects. Furthermore, upscaling PC1 significantly improves SORT performance in the 8B model, with a similar but (after correction) non-significant trend in 70B. This confirms that the observed behavioral changes are driven specifically by the causal direction of PC1 rather than nonspecific perturbations.
>
> ### **Figure 3: Grouped Query Attention.**
> The apparent mismatch comes from grouped query attention: in these models, multiple query heads share a single KV head. We will mention this in the figure description to prevent confusion.
>
> ### **LLM vs. human experimental setup.**
> The models were evaluated with independent questions, i.e. one encoding prompt with one retrieval query, rather than a sequence of multiple questions. We agree that this is an important deviation from the human protocol and has implications. We will state this and suggest for future work that seeks to do cognitive modeling to investigate these mechanisms.
>
> ### **Can secondary temporal information have identical enc. and ret. temporal readout directions?**
> Secondary temporal information can support retrieval-time temporal information, but it should generally fail to support both retrieval- and encoding-time temporal geometry along the same axis. The reason is that the downstream head has only one shared $W_V$. If its retrieval-time temporal signal is inherited from the primary head’s written time-coding direction $w^\*$, then that same projection must also act on $w^\*$ during encoding. But across encoding positions, the signal along $w^\*$ is not a clean monotonic code of current position; it is a non-monotonic sequence of reinstated past times. So the downstream head may look temporal at retrieval, yet its own value cache will not look temporally ordered along that same direction at encoding.
>
> ### **Specificity to long contexts?**
> New experiments on short contexts suggest that the same mechanism is still causally relevant (significant accuracy drop after ablating L8H11 in Llama3.1-8B for 1k word excerpts from Crime and Punishment (SORT), and for 2.5k random word sequences (SORT). We also find the same for 500 and 25k word excerpts from Picture of Dorian Gray in a new 4-segment ordering task.
>
> ### **Why is the top reinstatement score lower in the larger model?**
> L8H11 in Llama 3.1-8B places about 64% of its attention mass on the encoding-time span of the target segment, whereas L14H21 in the 70B model places about 39% there, which could explain the higher scores for the 8B model.
>
> ### **Demographic questions and audiobook listeners.**
> English proficiency and number of days since finishing the book were used for our current analyses and the remaining demographic questions were collected for future stratified analyses. Audiobook listeners were included. We will clarify this.
>
> Finally, we want to reiterate that we agree that the paper is strongest when it makes a careful distinction between:
> - arguing for a potential cognitive model (which only serves to motivate our research), and
> - a mechanistic investigation into temporal reinstatement circuitry in long-context transformers (what we provide).
>
> We will make sure that this distinction is clear.

---

> > ### Author Rebuttal · Reviewer_wUZw · 2026-03-31
> >
> > I thank the authors for addressing all my concerns. I will update my score accordingly.

---

### Official Review · Reviewer_wUNU · 2026-03-12

**Soundness:** 3
**Presentation:** 3
**Significance:** 3
**Originality:** 3
**Overall Recommendation:** 4
**Confidence:** 3

**Summary:**

This paper investigates the computational mechanisms underlying episodic-like temporal order memory in long-context Large Language Models (LLMs). The authors introduce the Sequence Order Recall Task (SORT) and curate a novel human behavioral dataset based on the full-length novel The Murder of Roger Ackroyd. They demonstrate a striking alignment between human participants and Llama 3.1 models (8B and 70B), both exhibiting a strong distance effect in their accuracy. To rule out reliance on narrative inference, the authors perturb causal coherence via sentence-block shuffling, revealing that models robustly maintain their temporal ordering capability.

Through mechanistic interpretability techniques (PCA and linear probing on KV-cache representations), the study uncovers that temporal context is stored as a low-dimensional code. Remarkably, this temporal context is reinstated during the retrieval phase almost entirely by a single specialized attention head (e.g., L8H11 in the 8B model and L14H21 in the 70B model). Causal interventions—projecting out or amplifying the temporal readout direction in these specific heads—drastically degrade or improve task performance, validating the causal necessity of this specific mechanism.

**Compliance With Llm Reviewing Policy:**

Affirmed.

**Key Questions For Authors:**

1.	Cross-Model Generalization: The discovery of a single attention head dominating temporal reinstatement is interesting. Is this specific to Llama 3? Could the authors provide preliminary qualitative results on at least one other open-weight long-context model (e.g., Qwen, Mistral, or a non-RoPE model) to demonstrate the universality of this mechanism?
2.	Multi-Segment Ordering: The current SORT benchmark elegantly isolates order judgments but relies on a simple binary choice between two segments. Real-world episodic memory and narrative comprehension often require reconstructing the chronological order of a sequence of events. Does this "single-head time reinstatement" mechanism scale to more complex ordering tasks, such as sorting multiple (e.g., 3 to 5) segments chronologically?
3.	Data Contamination Control: The Murder of Roger Ackroyd is in the public domain and likely exists in the pre-training corpus. While the sentence-shuffling experiment is a solid mitigation, could the authors verify if this single-head reinstatement phenomenon holds consistently on a completely synthetic, LLM-generated long narrative that the model could never have memorized?

**Limitations:**

yes

**Strengths And Weaknesses:**

Strengths:

•	Originality: The connection drawn between human cognitive episodic memory and LLM mechanistic interpretability is highly creative. The discovery that time reinstatement is localized to a single attention head in such large models is a surprising and clean finding.

•	Soundness: The experimental design is rigorous. The use of sentence-block shuffling elegantly isolates temporal tracking from narrative reasoning. Furthermore, the causal intervention experiments (ablating and amplifying the PC1 direction) form a robust logical loop that elevates the findings from mere correlation to causal mechanisms.

•	Significance: As the context window of LLMs continues to grow, understanding how models organize and retrieve temporal sequences is of paramount importance. This paper provides a stellar methodological template for localizing episodic-like circuits in transformers.

Weaknesses:

•	Generalization across models: The analysis is exclusively constrained to the Llama 3.1 family (8B and 70B). It remains an open question whether this "single-head time reinstatement" is an artifact of the Llama 3 architecture or a universal phenomenon across causal transformers.

•	Generalization to complex sequences: The conclusions heavily rely on the binary Sequence Order Recall Task (SORT), which only asks the model to compare the relative order of two 50-word segments. While this elegantly isolates basic order judgments, real-world episodic memory often involves reconstructing the timeline of multiple events. It is unclear if this single-head reinstatement mechanism scales to more complex ordering tasks, such as sorting multiple (e.g., 3 to 5) segments chronologically, or if the model relies on this exact temporal axis for downstream tasks that implicitly require multi-segment ordering.

---

> ### Author Rebuttal · Authors · 2026-03-31
>
> Thank you for the encouraging review and for highlighting both the originality of the human/LLM connection and the strength of the causal intervention loop. We agree on the important open questions concerning generalization: across model families, beyond binary comparisons, and beyond a public-domain novel.
> ### **Cross-model generalization beyond Llama 3.1.**
> To address this, we ran additional experiments on two non-Llama model families using shorter-context datasets matched to their context constraints (we use 8k token contexts datasets for training/TR-identification and causal interventions): Mistral v0.2-7B and Qwen2.5-7B. In both cases, we observe the same qualitative pattern as in the Llama models: a clear outlier head under our temporal reinstatement score (L7H18 with 0.53 for Mistral at a gap to the nearest score of 0.22; L14H0 with 0.4 for Qwen and a gap of 0.13), along with a causal dependence of task performance on that head (drop from 0.6 to 0.5 in Mistral and from 0.65 to 0.6 in the Qwen model after head-ablation). These results suggest that the “time-reinstatement head” phenomenon is not unique to Llama 3.1 and that it can still be localized with less data at shorter context.
> ### **Generalization to multi-segment ordering.**
> We agree that binary SORT isolates a clean primitive, but episodic behavior can also involve ordering multiple events. To test this, we created a new 4-segment ordering task and evaluated the 8B model on 500 word and 25k word excerpts. As in SORT, the model first processes an excerpt without any explicit indication of a temporal-ordering task; it is then asked to recover the chronological order of four 50-word segments, with labels (A, B, C, D). For each sample, we evaluate the log-probabilities of all 24 (4!) possible orderings and take the most probable ordering as the model’s response. On 1,000 samples with a 500-word excerpt context randomly sampled from The Picture of Dorian Gray, Llama 3.1-8B reaches a baseline mean rank-correlation of 0.63. Ablating L8H11 changes performance to 0.54 with a 95% CI of [0.51, 0.56], while ablations of other tested heads did not produce a comparable significant drop. With 25k word excerpts, we find a similar pattern: Rank correlation drops from 0.31 down to 0.11 when L8H11 is ablated, while we did not observe a drop in performance for other heads.
> We see this as suggestive evidence that the identified mechanism is relevant not only for binary comparisons, but also for more structured multi-segment temporal reconstruction. It additionally suggests that the mechanism plays a causal role at substantially shorter context lengths, though short-context ordering may additionally recruit more distributed circuitry.
> ### **Data contamination and memorization.**
> We agree that contamination is an important concern for any public-domain text. We note, however, that several parts of our evidence already guarantee that the main mechanistic finding is not simply an artifact of memorization. First, task performance is largely preserved on multiple sentence-block-shuffled versions of the novel, which are highly unlikely to have appeared in pretraining in that form. Second, the temporal readout directions are not fit on the book text: they are fit on a random-word sequence and validated on held-out documents that were not in the training data, including shuffled variants of the book text and a recent House of Commons transcript. This makes the localization result difficult to explain as memorization of the original book. To further support this, we now evaluated the Llama3.1 8B model on a synthetic story (90k tokens). Using this data, we find the same qualitative intervention effects (a robust positive slope seen in Figure 7).
>
>
> We will make clearer that our main claim is the mechanistic localization and causal validation of temporal reinstatement, while the new cross-model, multi-segment, and synthetic-narrative results provide preliminary evidence that the phenomenon is not limited to one model family, one task format, or depends on memorization during pretraining.

---

> > ### Author Rebuttal · Reviewer_wUNU · 2026-04-07
> >
> > Thank you for the detailed rebuttal. I have read the authors’ response carefully, and my main concerns have been adequately addressed. The clarifications and additional evidence strengthen the paper and improve my confidence in the technical soundness of the work. Based on the rebuttal, I maintain my positive assessment of the paper.

---

> > > ### Author Response · Authors · 2026-04-07
> > >
> > > Thank you for reading the rebuttal carefully and for the thoughtful acknowledgement. We appreciate your note that your main concerns have been adequately addressed and that the additional analyses improved your confidence in the technical soundness of the paper.
> > >
> > > We are especially glad that the new evidence on cross-model generalization, multi-segment ordering, and synthetic long narratives was helpful in resolving the concerns you expressed in the original review.
> > >
> > > If you feel that the rebuttal has strengthened the paper relative to your initial review, we would greatly appreciate it if you would consider updating your score to reflect your revised assessment.
> > >
> > > Thank you again for your time and for helping us strengthen this work.

---

### Official Review · Reviewer_hrtf · 2026-03-13

**Soundness:** 4
**Presentation:** 3
**Significance:** 3
**Originality:** 3
**Overall Recommendation:** 5
**Confidence:** 4

**Summary:**

In this paper, the authors propose a temporal order task for LLMs and use mechanistic interpretability methods to localize temporal information and identify attention heads responsible for specific temporal contexts. The Sequence Order Recall Task (SORT) presented consists of an encoding-retrieval paradigm in which models and human participants encode/read a novel or other large document and then judge the temporal order of two 50 word segments presented in a retrieval phase. Llama 3.1-8B and -70B models were evaluated on this task, with judgement performance compared to that of human participants, showing similar trends in judgement accuracy across segment distances. In order to assess if judgement accuracy was dependent on semantic, narrative structure, the authors conduct an additional evaluation in which the queried segments were shuffled during presentation, demonstrating that the effect of segment distance on judgement accuracy is preserved under shuffling.

In addition to the SORT dataset and human-performance comparison results, the work also details localization of a temporal code using mechanistic interpretation of the value cache of attention heads. They define a temporal reinstatement score (TRS) that measures the temporal alignment of a temporal read-out direction (i.e.  top principal component of the heads' value cache) with representative matrices constructed from the encoding and retrieval sequences. Analyzing this temporal information, it is determined that a single head within Llama 8b and 70b is driving judgement accuracy, regardless of input shuffling. The authors further confirm this determination by applying interventions, projecting out this temporal principal direction or increasing the gain on only the top reinstatement head, resulting in a decrease and increase in judgment accuracy, respectively.

**Compliance With Llm Reviewing Policy:**

Affirmed.

**Final Justification:**

I believe the authors have answered all questions satisfactorily and have presented significant and sound work.

**Key Questions For Authors:**

1. Could you expand a little more on the results seen in Figure 2? The largest LLM shows what appears to be a significant increase in judgement accuracy at distances of 16-24K. Is there narrative coherence explanation for this?

2. You indicate that human participants and the Llama 3.1-8B had statistically similar performance on the SORT benchmark. Are there any other aspects of their behavior on the benchmark that aligns? For instance, do human participants and the Llama model fail on similar segment pairs?

3. You mention you analyzed temporal order information of the Llama 3.1-8B model on random token sequences with and without RoPE enabled. Could you perhaps evaluate the Llama models on your SORT benchmark without RoPE enabled as well? Showing that judgement accuracy is maintained with positional embeddings would provide strong evidence toward them being unnecessary for this temporal reinstatement.

**Limitations:**

Yes.

**Strengths And Weaknesses:**

**Soundness:**

The work is technically sound and firmly rooted in prior work in human memory. The localization of the temporal reinstatement head is well-supported by the following intervention experiments, which corroborate the main finding that this is the primary driver of the model's judgement accuracy. The further analysis suggesting this result is not merely due to RoPE embeddings also provides substantial evidence for the authors' identification of an episodic reinstatement circuit. In terms of the dataset proposed, it seems unclear that judgement accuracy being a function of segment distance is enough to confirm the model's behavior is human-like in any meaningful way. The preservation of performance under input shuffling motivates the later exploration of attention head temporal reinstatement, but it is unclear if this meant to align with human participants.

**Presentation:**

The motivation behind the mechanistic interpretability is very well motivated and explored. However, the analysis providing evidence that positional encoding is not responsible for the temporal reinstatement effect presented here seems too important to be relegated to supplementary material. Additionally, the human participant results seem almost tangential to the core of the work.

**Significance:**

This work does further the fundamental understanding of how LLMs make use of temporal information within their contexts. The result showing that this ability is irrespective of (and potentially even undermined by) positional embedding is particularly interesting, and could have implications for future LLM design. Given that there are a number of long-context datasets/benchmarks already available, it is less clear if the SORT benchmark will have wider impact.

**Originality:**

This work incorporates a fresh perspective that sheds light on an aspect of LLMs' temporal reasoning ability and the approach is well-articulated.

---

> ### Author Rebuttal · Authors · 2026-03-31
>
> Thank you for the positive assessment and for the thoughtful questions! We are glad that you found the mechanistic localization and intervention results convincing and significant. We agree that the paper will benefit from an expanded discussion of the behavioral findings, and from integrating the findings on the role of RoPE into the main text.
>
> ### **Narrative coherence and the 16k–24k bin in Llama 3.1-70B.**
>
> To investigate the higher performance of the 70B model in the 16k–24k distance bin, we ran an additional exploratory post-hoc annotation analysis of the queried segment pairs. For each pair, we presented the two segments in their true order and asked an external LLM annotator (with reasoning) to judge whether the pair contained a clear narrative connection that could explain the ordering (e.g., temporal progression, dialogue continuation, revelation, state continuity), with labels present / absent / unclear. Using this annotation as a proxy for narrative/causal connectedness, we found that within the 16k–24k bin, Llama 3.1-70B accuracy is not positively associated with narrative coherence; surprisingly, the correlation is negative (r = -0.35, p = 0.035). While we view this as exploratory rather than definitive (future work should additionally employ human annotations; and ideally larger scale data), it argues against an obvious “narrative coherence” explanation for the bump in accuracy.
>
> ### **Behavioral alignment beyond the distance effect.**
>
> We agree that our current evidence supports a qualitative alignment in the distance effect, not a broader claim of item-level human-like behavior. After conditioning on distance, the residual alignment between model and human responses appears substantially weaker. We therefore plan to revise the paper to make this scope explicit: the human comparison is intended as a behavioral context for the task, while the main contribution is the mechanistic localization of temporal reinstatement. We will also clarify an important protocol difference: humans answered multiple queries after naturalistic reading (~90% of segment pairs were judged by human participants after they had already judged at least one pair), whereas the models were evaluated with independent long-context prompts. This difference may matter for fine-grained item-level agreement and should be stated more clearly as a limitation.
>
> ### **Clarification regarding shuffled narrative results.**
>
> We do not intend to model human behavioral data with the shuffled versions of the task and will make this explicit in the text. The finding that models show the same behavioral pattern on shuffled narratives is meant to rule out that the model depends on temporal reasoning about the narrative structure rather than on mechanisms of reinstatement.
>
> ### **Highlighting RoPE results.**
>
> We agree that the Appendix A.1 result is an interesting finding that is aligned with results from other recent research and should not be easy to miss. In the revision, we will summarize it directly in the main paper and connect it to the mechanistic findings. Our intent in the current version was to focus on the retrieval/reinstatement mechanism, how temporal information is reinstated and causally used at test time, rather than on the encoding-phase origin of positional information. But we agree that the current placement understates the interest in this result.
>
> ### **Why we do not evaluate SORT with RoPE simply switched off.**
>
> Turning off RoPE at inference is known to produce severe distribution shifts and substantially degrades the model's overall functioning, so any task-level drop would conflate the specific issue of temporal coding with a much broader out-of-distribution failure. For that reason, we prefer the representational control reported in Appendix A.1, which focuses merely on the encoding side and isolates whether the encoded temporal signal studied here depends on explicit positional encoding. We will make this rationale explicit in the revision.
>
> ### **Presentation of the human results.**
>
> We also agree that the current paper can better separate the roles of the two empirical components: the human data establish a useful naturalistic benchmark and motivate the temporal-order memory phenomenon, while the core scientific contribution is the identification and causal validation of a reinstatement mechanism in the model. We will make this clear in the introduction and discussion.

---

> > ### Author Rebuttal · Reviewer_hrtf · 2026-04-03
> >
> > I appreciate the authors' response and additional analysis, which suitably answers my questions. The negative correlation between accuracy and narrative coherence is also a very unexpected outcome that could form the basis of some very interesting future work.

---

### Decision · Program_Chairs · 2026-04-30

**Decision:**

Accept (regular)

**Comment:**

Based on the reviews, the work is technically sound and well grounded in prior research on human memory, and it offers a creative connection between episodic memory and mechanistic interpretability in LLMs. The identification of a temporal reinstatement attention head is strongly supported by causal intervention experiments, and further reinforced by evidence that the effect is not simply driven by positional embeddings. The experimental design is rigorous, with careful controls such as shuffling and probing methods that effectively isolate temporal reasoning. However, there are concerns that the dataset and evaluation do not clearly demonstrate that the model behavior is meaningfully human like, and the human participant results appear somewhat peripheral to the main contribution. In addition, important analyses, such as ruling out positional encoding effects, are relegated to the supplementary material despite their significance. While the work advances understanding of how LLMs process temporal information and provides a useful methodological contribution, the broader impact of the proposed benchmark remains somewhat uncertain given existing alternatives. Overall, I support acceptance of the paper.